# Accurate cross-species 5mC detection for Oxford Nanopore sequencing in plants with DeepPlant

He-Xu Chen[1,2,7], Zhen-Dong Liu[3,7], Xin Bai[1,7], Bo Wu[1,7], Rong Song[1,7], Hui-Cong Yao[2], Ying Chen[1], Wei Chi[4] ✉, Qian Hua [5] ✉, Liang Cheng [6] ✉ & Chuan-Le Xiao [1] ✉

Nanopore sequencing enables comprehensive detection of 5-methylcytosine (5mC), particularly in repeat regions. However, CHH methylation detection in plants is limited by the scarcity of high-methylation positive samples, reducing generalization across species. Dorado, the only tool for plant 5mC detection on the R10.4 platform, lacks extensive species testing. Here, we develop DeepPlant, a deep learning model incorporating both Bi-LSTM and Transformer architectures, which significantly improves CHH detection accuracy and performs well for CpG and CHG motifs. We address the scarcity of methylation-positive CHH training samples through screening species with abundant high-methylation CHH sites using bisulfite-sequencing and generate datasets that cover diverse 9-mer motifs for training and testing DeepPlant. Evaluated across nine species, DeepPlant achieves high whole-genome methylation frequency correlations (0.705-0.838) with BS-seq data on CHH, improved by 23.4-117.6% compared to Dorado. DeepPlant also demonstrates superior single-molecule accuracy and F1 score, offering strong generalization for plant epigenetics research.

DNA methylation, specifically 5-methylcytosine (5mC), is an essential epigenetic modification regulating numerous biological processes in plants[1], such as gene expression[2], transposon silencing[3], and genome stability[4,5]. Unlike in animals, where 5mC primarily occurs at CpG sites, plant 5mC exits across three different sequence contexts, CpG, CHG, and CHH (where H represents A, T, or C). CHH methylation, though less abundant, plays a critical role in silencing transposable elements (TEs), which is essential for maintaining genome integrity during plant development and stress responses[3].

Several methods have been developed for detecting 5mC[6,7], with bisulfite sequencing (BS-seq)[8] being the most widely used for all three methylation contexts. However, BS-seq's reliance on short-read sequencing technologies limits its ability to accurately profile complex and repetitive genomic regions, such as centromeres and transposable elements (TEs)[9]. Additionally, BS-seq introduces biases, such as DNA damage[10], which can impair accuracy-particularly for CHH motifs that only have half effective coverage due to being asymmetric between forward and reverse DNA strands. Recent advancements in third-generation sequencing, particularly with Oxford Nanopore Technologies (ONT), present a promising alternative. Nanopore sequencing signals can be directly utilized to detect DNA modifications on native long reads[11–15], enabling more comprehensive analysis

[1]State Key Laboratory of Ophthalmology, Zhongshan Ophthalmic Center, Sun Yat-sen University, Guangdong Provincial Key Laboratory of Ophthalmology and Visual Science, Guangzhou, China. [2]School of Artificial Intelligence, Sun Yat-Sen University, Zhuhai, China. [3]School of Computer and Information Engineering, Shanghai Polytechnic University, Shanghai, China. [4]Shenzhen Eye Hospital, Shenzhen Eye Medical Center, Southern Medical University, Shenzhen, China. [5]School of Life Science, Beijing University of Chinese Medicine, Beijing, China. [6]College of Bioinformatics Science and Technology, Harbin Medical University, Harbin, China. [7]These authors contributed equally: He-Xu Chen, Zhen-Dong Liu, Xin Bai, Bo Wu, Rong Song.
✉e-mail: chiwei@mail.sysu.edu.cn; huaq@bucm.edu.cn; liangcheng@hrbmu.edu.cn; xiaochuanle@126.com

of genomic regions. A key breakthrough in this area for plant has been DeepSignal-Plant[16], our deep learning tool developed for genome-wide 5mC detection in CpG, CHG, and CHH contexts. Trained on *Arabidopsis thaliana* and *Oryza sativa*, DeepSignal-Plant demonstrated high accuracy across these contexts, correlating strongly with BS-seq while providing enhanced methylation information in repetitive regions.

The recently released ONT's R10.4 FlowCell has dramatically improved accuracy and stability in basecalling compared to earlier versions and has gradually become the mainstream product. However, software compatible with the R10.4 FlowCell has lagged behind. Substantial changes in the nanopore protein structure and signal collection frequency of R10.4 have led to differences in electrical signal output and data storage formats. As a result, methylation detection tools developed for the R9.4 platform, such as Tombo[17], Megalodon[18], and DeepSignal-Plant[16], cannot be directly applied to the R10.4 FlowCell. To address the compatibility issue with the R10.4 FlowCell, ONT has introduced new 5mC detection models for their Dorado[19] software. While Dorado performs well in detecting high-methylation levels of CpG and CHG in plants on R10.4, it struggles with CHH methylation detection (see Results), most likely due to the limited availability of positive CHH samples for training.

In this work, we analyze publicly available BS-seq datasets and screen species with abundant high-methylation CHH sites for model training. We generate new Nanopore and BS-seq data from selected species, significantly increasing the number of CHH-positive samples. The new dataset covers 97.2% of all possible 9-mers centered with CHH motifs. In parallel, we develop DeepPlant, which outperforms Dorado, achieving a correlation improvement ranging from 0.135 to 0.381 with BS-seq in whole-genome CHH methylation frequency quantification. DeepPlant also demonstrates superior single-molecule accuracy, F1 score, and recall, while maintaining greater stability across all tested species. These results suggest that DeepPlant has strong generalizability and holds significant potential for broad applications in plant methylation detection.

## Results

### Sample selection for generalized CHH methylation model training

This study aimed to enhance nanopore-based 5mC detection across plant species, with a particular focus on developing a CHH methylation model that generalizes well across species. A critical challenge in this process was obtaining methylation-positive samples with diverse motif contexts. In a previous study, CHH-positive samples were sourced from genomic CHH sites with high-methylation levels (≥90%) based on BS-seq data[16]. However, collecting samples with high CHH methylation levels and broad k-mer coverage is difficult due to the generally low CHH methylation content (-1–17% reported in 34 angiosperms)[20] in most plants. And highly methylated CHH sites represent only about 0.02–0.12% of BS-seq quantified CHH sites in *A. thaliana* and *O. sativa* (Supplementary Data 1) used for training DeepSignal-Plant[16].

To collect more representative positive training samples, we reviewed existing literature[20,21] and analyzed the abundance and context k-mer diversity of high-methylation CHH motifs using previously published BS-seq data from 10 plant species[16,22–35] (Supplementary Data 1). These species included *Arabidopsis thaliana* (a maximum of 0.03% high-methylation CHH sites among tested datasets), *Oryza sativa* (0.12%), *Beta vulgaris* (1.27%), *Salvia miltiorrhiza* (2.78%), *Solanum tuberosum* (1.96%), *Ricinus communis* (3.91%), *Citrus sinensis* (1.35%), *Gossypium hirsutum* (0.01%), *Solanum lycopersicum* (0.78%), and *Physcomitrium patens* (7.28%). The samples with the highest ratios of high-methylation CHH sites or the greatest k-mer context diversity were *S. tuberosum* tuber, *S. miltiorrhiza* root, *P. patens* gametophore, *R. communis* embryo, *S. lycopersicum* fruit, *C. sinensis* fruit pericarp, and *B. vulgaris* leaves (Fig. 1a and Supplementary Data 1).

We then collected tissue samples from seven of these species and conducted BS-seq. For better sample diversity, *A. thaliana*, *O. sativa*, *Glycine max*, *Vitis vinifera*, and *Marchantia polymorpha* were also added to the analysis. Among these, BS-seq data from *S. miltiorrhiza* root provided the largest number of CHH-positive samples, followed by *R. communis* embryo and *S. tuberosum* tuber (Fig. 1b, c). The same three DNA samples were therefore subjected to nanopore sequencing on R10.4 platform. Low-mapping rate (<5%) of *P. patens* and *M. polymorpha* BS-seq suggested sample impurity, and the *G. max* strain showed a relatively high nucleotide difference level to the reference genome (GCF_000004515.6), leading them to be abandoned in further analysis. Analysis of the nanopore data revealed that *S. miltiorrhiza* alone covered 93.4% of all possible 9-mer CHH contexts. However, the number of high-methylation sites for specific CHH motifs, such as CCA, CCC, and CCT, was low across all species (Fig. 1c). Consequently, we combined the datasets from *S. miltiorrhiza*, *S. tuberosum*, and *R. communis*, resulting in 97.2% coverage of all possible 9-mer contexts centered with CHH motifs, with an average of 9225 samples per context.

To evaluate the model's generalizability, we selected another six species with varying CHH methylation levels for testing, which are *A. thaliana* (3.01% overall CHH methylation ratio), *V. vinifera* (8.04%), *O. sativa* (3.50%), *B. vulgaris* (10.23%), *S. lycopersicum* (15.10%), and *C. sinensis* (10.61%) (Supplementary Data 2). The number of high-methylation CHH sites in their BS-seq datasets also differed from 2047 in *A. thaliana* to 520,150 in *S. lycopersicum*. These results indicate that our selected datasets provide a broad and representative foundation for both model training and evaluation.

### Deep neural network architectures and model training for plant 5mC detection

Based on the datasets collected, we developed DeepPlant, a deep learning tool for accurate 5mC detection in plants. DeepPlant employs a triple-encoder architecture (Fig. 2a), similar to our previous tools, DeepSignal[36] and DeepBam[36]. This architecture includes separate encoders for k-mer sequence information (from Dorado basecalling), raw signal features, and a secondary collaborative encoding. Systematic ablation analysis showed that discarding any of the encoders (sequence encoder, signal encoder, and combine encoder) would significantly impact the model's performance (Supplementary Data 3). A classifier then determines the methylation status of cytosines located at the center of the k-mer. Two deep neural network architectures were implemented in DeepPlant-one based on Bidirectional Recurrent Neural Networks with LSTM units (Bi-LSTM) and the other using Transformer encoders, forming a BERT-like network (Fig. 2a–c and Supplementary Data 4).

Cytosine methylation can affect the signals of adjacent sequences, so the input feature length influences model performance[16,37]. To assess the impact of using different k-mer lengths, we trained DeepPlant models using 9-mer, 13-mer, and 51-mer samples. Testing on *O. sativa* and *A. thaliana* showed that models trained with 9-mer samples, matching Dorado's input length, significantly outperformed Dorado, suggesting the robustness of our training dataset. Among the models trained using different k-mer lengths, the 51-mer model had the highest accuracy on randomly sampled testing dataset (Fig. 2d); however, it is overfitted since it led to poorer methylation frequency quantifications (Fig. 2e). Overall, the 13-mer Bi-LSTM model (default DeepPlant model) was regarded optimal and further assessed in following sections.

In addition, we trained CHG and CpG detection models using a similar Bi-LSTM architecture, which are detailed in the Methods section and Supplementary Fig. 1a–d.

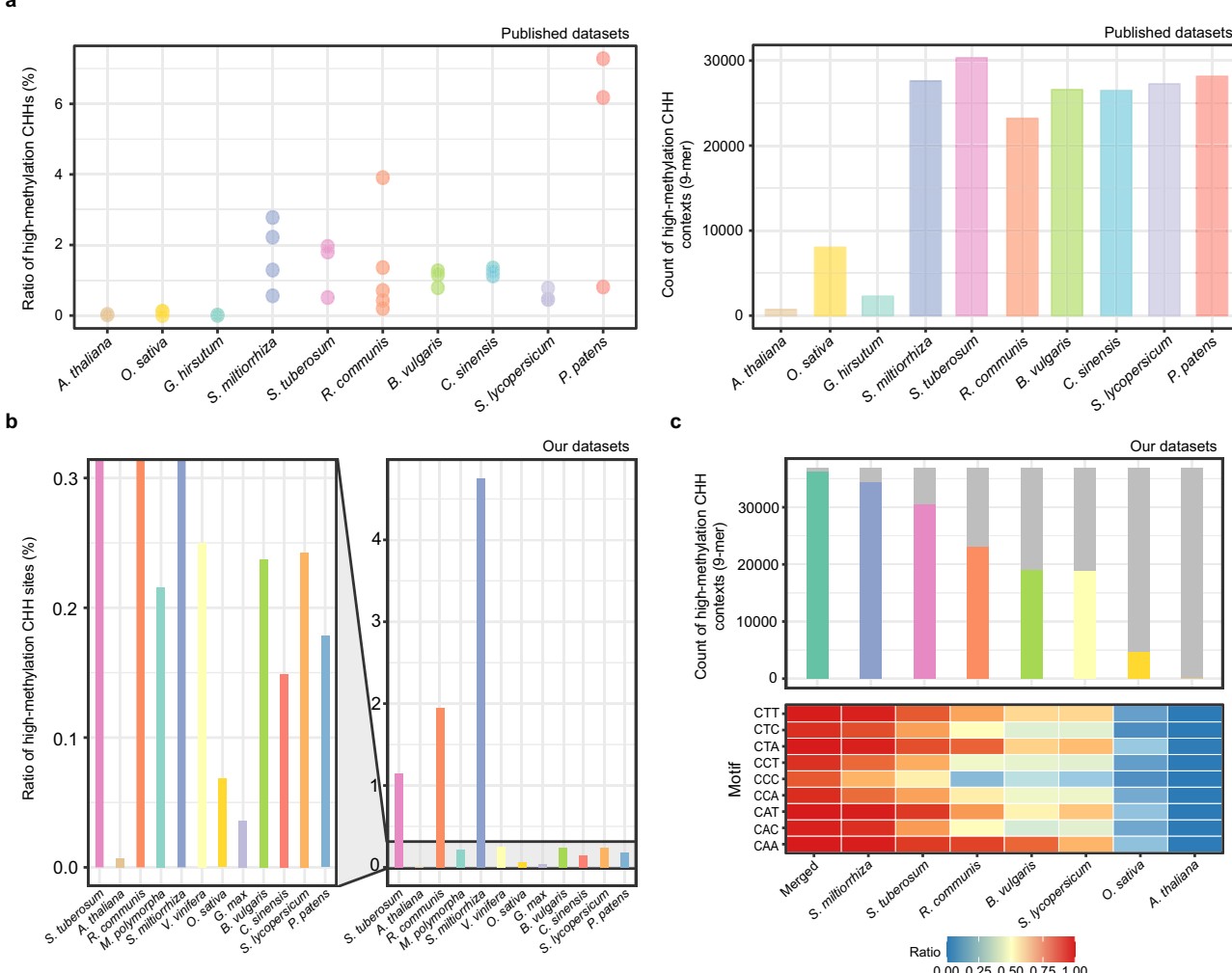

**Fig. 1 | Selection of plant samples for CHH methylation training feature collection.** CHH methylation sites, particularly those with high-methylation levels (≥90%), are rare in plants. This figure presents the statistics on high-methylation CHH sites from previously published bisulfite sequencing (BS-seq) datasets[16,22–35] and those generated in this study. **a** Ratios of high-methylation CHH sites among quantified CHH motifs (≥5 read coverage) (left panel) and the number of covered 9-mer contexts (right panel) in BS-seq datasets from ten plant species. Only 9-mers observed at three or more high-methylation CHH sites were considered. Species other than *A. thaliana* and *O. sativa* were selected based on high CHH methylation ratios reported in previous studies[20,21]. **b** Ratios of high-methylation CHH sites in BS-seq datasets sequenced for this study, including *A. thaliana*, *O. sativa*, and species with abundant high-methylation CHH sites identified in (**a**), as well as *Glycine max* and *Marchantia polymorpha*. **c** Number of covered 9-mer contexts (top) and heatmap of context abundance (bottom) grouped by CHH motifs in nanopore datasets from six plant species. In the top panel, the top line of each bar corresponds to the number (36,864) of all possible 9-mer sequences centered with a CHH motif. A 9-mer was considered covered if present in 50 or more positive training samples and had at least an equal number of negative samples. "Mixed" refers to combined samples from *S. miltiorrhiza*, *R. communis*, and *S. tuberosum*. Source data are provided as a Source Data file.

## Evaluation of 5mC methylation frequency quantification by DeepPlant

Aggregated methylation frequencies-defined as the ratio of methylated reads to total local effective reads-at CHH sites across the genome is a key metric for evaluating the performance of methylation detection models. As shown in Fig. 2c, d, this metric does not always positively correlate with single-molecule performance but can indicate a model's overall effectiveness. To assess the quantitative performance of DeepPlant, we profiled CHH methylation frequencies across three training datasets (*S. miltiorrhiza*, *S. tuberosum*, *R. communis*) and six testing datasets: *B. vulgaris*, *O. sativa*, *A. thaliana*, *V. vinifera*, *C. sinensis*, and *S. lycopersicum*. BS-seq data were also used to profile methylation frequencies at each cytosine site, serving as a control to evaluate the performance of nanopore-based methylation callers.

To minimize the sum of false-positive and false-negative calls, we applied an adaptive methylation score threshold selection method in DeepPlant (Supplementary Fig. 2; see "Methods"). We also applied the same approach in methylation frequency profiling using Dorado calls, and it improved the site-level methylation quantification correlations with BS-seq for Dorado compared to using its default settings (Table 1). On the six testing datasets, at a sequencing depth of 30×, DeepPlant achieved 0.705–0.838 Pearson's correlations (*r*) with BS-seq (Table 1 and Supplementary Data 5). For species with relatively high CHH methylation content, such as *B. vulgaris*, *C. sinensis*, and *S. lycopersicum*, the correlations exceeded 0.80. Across all testing datasets, DeepPlant consistently showed higher correlations and more similar methylation patterns (Supplementary Fig. 3a–c) with BS-seq compared to Dorado (using its default settings), with advantages ranging from 0.078 to 0.324 at 10× and from 0.135 to 0.381 at 30× sequencing depth (Table 1), even higher than the advantages observed on the three training datasets. As nanopore sequencing depth increased, the correlation between DeepPlant's aggregated methylation frequencies and BS-seq results steadily improved, while Dorado showed reduced correlations with BS-seq in most cases as sequencing depth increased.

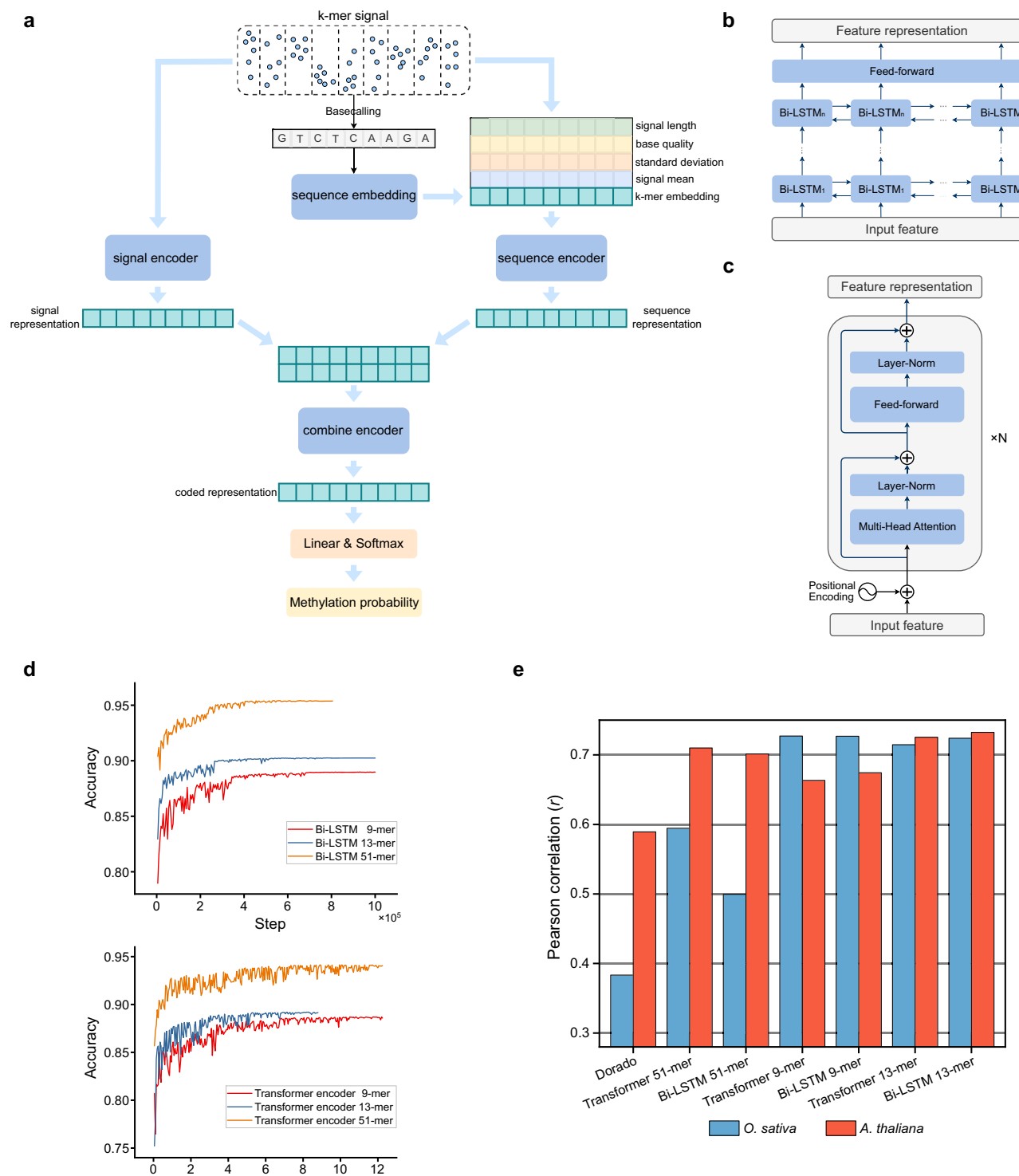

**Fig. 2 | Deep neural network architecture and model comparison. a** Overview of the signal features used by DeepPlant and the triple-encoder architecture. **b, c** Bi-LSTM and Transformer encoder architectures applied in DeepPlant. **d** Accuracy progression during CHH methylation detection training, comparing the performance of Bi-LSTM and Transformer encoders across different k-mer lengths. 9-, 13-, and 51-mer denote the lengths of model feature contexts surrounding target C at CHH sites. **e** Quantitative evaluation of CHH methylation detection accuracy by different models on single chromosomes using 43× *O. sativa* and 35× *A. thaliana* nanopore data. Pearson correlations were calculated between nanopore and corresponding BS-seq data. Source data are provided as a Source Data file.

Notably, DeepPlant achieved a nearly two-fold correlation coefficient with BS-seq on *O. sativa* compared to Dorado (0.654 vs. 0.329).

These results highlight DeepPlant's superior generalizability in CHH methylation frequency quantification. We also evaluated methylation frequency quantification using DeepPlant's CHG and CpG

models across the same nine datasets, and the results are detailed in Supplementary Data 6 and Supplementary Note 1. Overall, our CHG model outperformed Dorado across the datasets, though with a smaller margin compared to the CHH model. And the CpG model performed slightly better than Dorado on seven of nine tested datasets

**Table 1 | Quantitative evaluation of CHH methylation detection by DeepPlant and Dorado**

| Species | Tool | Sequencing depth | | | | | |
|---|---|---|---|---|---|---|---|
| | | 5× | 10× | 15× | 20× | 25× | 30× |
| **Training datasets** | | | | | | | |
| *S. miltiorrhiza* | DeepPlant | 0.8472 | 0.8524 | 0.8601 | 0.8695 | 0.8780 | – |
| | Dorado | 0.7604 | 0.7500 | 0.7443 | 0.7420 | 0.7441 | – |
| | Dorado* | 0.8069 | 0.7975 | 0.7890 | 0.7821 | 0.7784 | – |
| *S. tuberosum* | DeepPlant | 0.8048 | 0.8173 | 0.8251 | – | – | – |
| | Dorado | 0.6964 | 0.6899 | 0.6861 | – | – | – |
| | Dorado* | 0.7453 | 0.7412 | 0.7365 | – | – | – |
| *R. communis* | DeepPlant | 0.8340 | 0.8532 | 0.8592 | 0.8663 | 0.8742 | 0.8814 |
| | Dorado | 0.7849 | 0.7819 | 0.7752 | 0.7714 | 0.7713 | 0.7735 |
| | Dorado* | 0.8098 | 0.8133 | 0.8085 | 0.8048 | 0.8036 | 0.8041 |
| **Testing datasets** | | | | | | | |
| *A. thaliana* | DeepPlant | 0.6410 | 0.6611 | 0.6733 | 0.6853 | 0.6998 | 0.7139 |
| | Dorado | 0.5908 | 0.5828 | 0.5813 | 0.5795 | 0.5789 | 0.5787 |
| | Dorado* | 0.6322 | 0.6324 | 0.6365 | 0.6378 | 0.6394 | 0.6396 |
| *B. vulgaris* | DeepPlant | 0.7655 | 0.7740 | 0.7826 | 0.7920 | 0.8006 | 0.8074 |
| | Dorado | 0.6385 | 0.6242 | 0.6123 | 0.6062 | 0.6054 | 0.6051 |
| | Dorado* | 0.7253 | 0.7182 | 0.7076 | 0.6996 | 0.6937 | 0.6904 |
| *O. sativa* | DeepPlant | 0.6401 | 0.6535 | 0.6659 | 0.6796 | 0.6937 | 0.7051 |
| | Dorado | 0.3266 | 0.3292 | 0.3269 | 0.3247 | 0.3237 | 0.3240 |
| | Dorado* | 0.4833 | 0.4937 | 0.4860 | 0.4728 | 0.4605 | 0.4510 |
| *V. vinifera* | DeepPlant | 0.7268 | 0.7326 | 0.7445 | 0.7588 | 0.7719 | 0.7832 |
| | Dorado | 0.6499 | 0.6360 | 0.6284 | 0.6233 | 0.6201 | 0.6193 |
| | Dorado* | 0.7082 | 0.7026 | 0.7000 | 0.6976 | 0.6949 | 0.6928 |
| *C. sinensis* | DeepPlant | 0.7616 | 0.7800 | 0.7949 | 0.8064 | 0.8175 | 0.8259 |
| | Dorado | 0.6644 | 0.6595 | 0.6563 | 0.6544 | 0.6538 | 0.6533 |
| | Dorado* | 0.7165 | 0.7146 | 0.7090 | 0.7055 | 0.7032 | 0.7017 |
| *S. lycopersicum* | DeepPlant | 0.7808 | 0.7903 | 0.8022 | 0.8157 | 0.8277 | 0.8378 |
| | Dorado | 0.6759 | 0.6655 | 0.6569 | 0.6546 | 0.6546 | 0.6577 |
| | Dorado* | 0.7315 | 0.7260 | 0.7195 | 0.7153 | 0.7134 | 0.7133 |

Note: Each value represents the genome-wide Pearson correlation between the methylation callers and whole-genome BS-seq results at the corresponding sequencing depths of downsampled nanopore datasets. For each species, the same BS-seq dataset (Supplementary Data 2) was used for all tests. The results of applying DeepPlant's threshold selection method (Supplementary Fig. 2) to Dorado are labeled as "Dorado*".

and also performed slightly better than Rockfish on eight tested datasets.

## Single-molecule methylation detection performance of DeepPlant

Nanopore sequencing, as a single-molecule long-read sequencing technology, offers a distinct advantage in detecting methylation for individual molecules compared to BS-seq. To assess DeepPlant's performance in this context, we conducted a comprehensive analysis across nine datasets. Reference sites were selected using corresponding BS-seq data, focusing on fully methylated (100% methylation frequency) and unmethylated (0% methylation frequency) CHH sites, with a minimum coverage of 5×. Given the scarcity of fully methylated CHH sites in the analyzed species, the ratio of fully methylated to unmethylated CHH genomic sites (Supplementary Data 7) is significantly lowered compared to the realistic single-molecule methylated to unmethylated CHH motif ratios in the analyzed species' DNAs (Supplementary Data 2), where the single-molecule performance could be indirectly evaluated through above methylation frequency quantification assessments. Direct evaluations of DeepPlant and Dorado on the imbalanced fully methylated/unmethylated datasets mainly provided insights into the accuracy of unmethylated site detection, and DeepPlant outperformed Dorado in all instances (Supplementary Data 7). Recognizing the importance of both methylated and

unmethylated calls, we then compared the single-molecule performance of DeepPlant and Dorado on datasets with a balanced representation of fully methylated and unmethylated samples. On the training datasets, DeepPlant achieved F1 scores exceeding 0.9 for *S. miltiorrhiza* and *R. communis*, outperforming Dorado across all three species. Notably, the F1 score for *S. miltiorrhiza* was more than 10% higher than Dorado. Results on the testing datasets (Table 2 and Supplementary Data 8) demonstrated that DeepPlant consistently outperformed Dorado and achieved higher F1 scores across all six species, with notable gains of 6.8%, 5.94%, and 5.48% for *O. sativa*, *B. vulgaris*, and *S. lycopersicum*, respectively. DeepPlant maintained <6% false-positive rates (FPRs) across all testing and training datasets (Supplementary Fig. 4a–i). In contrast, Dorado exhibited significantly higher FPRs on *C. sinensis*, *B. vulgaris*, *O. sativa*, and *S. lycopersicum*, with rates of 24.1%, 11.1%, 10.1% and 11.0%, respectively.

Further analyses using Receiver Operating Characteristic (ROC) and Precision–Recall (PR) curves (Supplementary Fig. 5a–r) confirmed DeepPlant's advantage at the single-molecule level. Across testing datasets, the area under the ROC curve (AUC) of DeepPlant increased by 0.22–6.77%, and the area under the PR curve (AP) improved by 1.31–7.29% compared to Dorado. The advantages were more pronounced in the training datasets. It is important to note that these metrics were calculated based on CHH sites with extreme methylation frequency levels. In the previous section, we observed much greater

**Table 2 | Single-molecule evaluation of CHH methylation detection**

| Species | Tool | F1 score | Accuracy | Recall | Precision | AUC | AP |
|---|---|---|---|---|---|---|---|
| **Training datasets** | | | | | | | |
| *S. miltiorrhiza* | DeepPlant | 0.9051 | 0.9083 | 0.8745 | 0.9378 | 0.9514 | 0.9617 |
| | Dorado | 0.7920 | 0.8000 | 0.7613 | 0.8252 | 0.8655 | 0.8526 |
| *S. tuberosum* | DeepPlant | 0.8869 | 0.8925 | 0.8432 | 0.9355 | 0.9365 | 0.9518 |
| | Dorado | 0.8042 | 0.8076 | 0.7900 | 0.8188 | 0.8693 | 0.8695 |
| *R. communis* | DeepPlant | 0.9008 | 0.9064 | 0.8502 | 0.9578 | 0.9460 | 0.9595 |
| | Dorado | 0.8577 | 0.8694 | 0.7873 | 0.9419 | 0.9224 | 0.9332 |
| **Testing datasets** | | | | | | | |
| *A. thaliana* | DeepPlant | 0.7883 | 0.8202 | 0.6697 | 0.9579 | 0.8662 | 0.9039 |
| | Dorado | 0.7722 | 0.8050 | 0.6611 | 0.9283 | 0.8481 | 0.8800 |
| *B. vulgaris* | DeepPlant | 0.8682 | 0.8775 | 0.8072 | 0.9392 | 0.9192 | 0.9405 |
| | Dorado | 0.8088 | 0.8217 | 0.7544 | 0.8717 | 0.8907 | 0.8942 |
| *O. sativa* | DeepPlant | 0.8867 | 0.8932 | 0.8355 | 0.9446 | 0.9375 | 0.9540 |
| | Dorado | 0.8186 | 0.8309 | 0.7632 | 0.8828 | 0.8887 | 0.9006 |
| *V. vinifera* | DeepPlant | 0.8414 | 0.8566 | 0.7608 | 0.9411 | 0.8964 | 0.9227 |
| | Dorado | 0.8150 | 0.8349 | 0.7274 | 0.9267 | 0.8941 | 0.9096 |
| *C. sinensis* | DeepPlant | 0.8221 | 0.8433 | 0.7241 | 0.9508 | 0.9256 | 0.9375 |
| | Dorado | 0.7830 | 0.7788 | 0.7982 | 0.7684 | 0.8579 | 0.8646 |
| *S. lycopersicum* | DeepPlant | 0.8872 | 0.8946 | 0.8294 | 0.9538 | 0.9332 | 0.9510 |
| | Dorado | 0.8324 | 0.8407 | 0.7910 | 0.8784 | 0.8918 | 0.9061 |

Note: This table presents single-molecule methylation evaluation results of DeepPlant (13-mer model) and Dorado across different species. Corresponding ROC (Receiver Operating Characteristic) and PR (Precision–Recall) curves are provided in Supplementary Fig. 5. And AUC and AP denote the area under ROC curve and the area under PR curve, respectively.

advantages of DeepPlant over Dorado in overall methylation frequency quantification. These results suggest that the tested Dorado model could be overfitted to extreme CHH sites.

We also evaluated the CpG and CHG models of DeepPlant across the datasets. Though with smaller advantages than the CHH model, both CpG and CHG models of DeepPlant demonstrated better single-molecule performance compared to Dorado, and the CpG model outperformed Rockfish on most metrics, with detailed results presented in Supplementary Data 6 and Supplementary Note 2.

## CHH methylation profiling of *O. sativa* centromere and transposon regions by DeepPlant

Centromeres are crucial structures in eukaryotic chromosomes, playing essential roles in mitosis and meiosis[38]. In plants, they are predominantly composed of satellite repeats, transposable elements (TEs), and a small number of genes[39]. The highly repetitive nature of centromeric sequences presents significant challenges for accurate assembly and functional analysis, including the study of their methylation patterns. Despite the agricultural importance of *O. sativa*, the methylation characteristics of its centromeres have been largely unexplored. Leveraging ~43× O. sativa nanopore data with a read N50 of 12.8 kb, we conducted an in-depth analysis of centromeric methylation patterns using DeepPlant on the T2T-NIP[40] reference genome (Supplementary Data 9). DeepPlant almost completely profiled centromeric regions of chromosomes 4, 5, 8, and 12, while the largest gap in coverage was observed in chromosome 11 (Fig. 3a). Across non-centromeric regions, DeepPlant quantified methylation frequencies for ~98% of CHH sites, representing ~26% improvements compared to the results achieved with ~52× BS-seq data (Fig. 3b). In centromeric regions, DeepPlant covered 88.0% CHH sites, more than double the coverage ratio of BS-seq (37.7%) (Fig. 3b). CHH coverage by DeepPlant in centromeric regions showed only slight reduction compared to mean genome sequencing depth (39.3×/43×), whereas BS-seq exhibited a more pronounced decrease (22.4×/52×). Exemplary centromeric regions successfully profiled by DeepPlant but left blank by BS-seq are shown in Fig. 3c and

Supplementary Fig. 6a–c. Moreover, in the sub-telomeric region of chromosome 10, within the gene *AGIS_Os10g035850* (*LOC_Os10g43075* in IRGSP-1.0/MSU7), all 3830 CHH motifs were profiled by DeepPlant, compared to only 2197 motifs profiled by BS-seq.

DeepPlant's ability to quantify methylation states in a strand-specific manner was further demonstrated in the analysis of *O. sativa* centromeric TEs (from genome annotation of T2T-NIP). CHH methylation levels were significantly higher in TEs compared to centromeric protein-coding regions (Fig. 3d). Among seven different types of centromeric transposons, no significant strand bias was observed overall. However, when looked at independently, Ac/Ds and Mariner sub-class transposons exhibited higher methylation on the forward strand, while others, including LINE and Ty1-copia, showed higher methylation on the reverse strand (Fig. 3e). This result echoed a previous study that reported strand-biased methylation in *A. thaliana* centromeres[41]. These results demonstrate that the combination of DeepPlant and nanopore sequencing offers enhanced coverage and accuracy for profiling methylation in complex genomic regions, such as centromeres and transposable elements, compared to traditional BS-seq approaches.

## Discussion

In this study, we introduce DeepPlant, a deep learning tool designed to accurately detect 5-methylcytosine (5mC) modifications across all sequence contexts-CpG, CHG, and particularly CHH-in plant genomes using Oxford Nanopore Technologies (ONT) R10.4 sequencing data. By addressing the limitations of existing methylation detection methods, especially in the CHH context, DeepPlant significantly enhances our ability to profile plant epigenomes comprehensively, including complex and repetitive genomic regions.

A critical challenge in CHH methylation detection has been the scarcity of high-methylation CHH sites for collecting positive samples, which hampers model training and generalization across species. Researchers have traditionally employed in vitro DNA methylation enzyme reactions to provide positive samples for CpG methylation

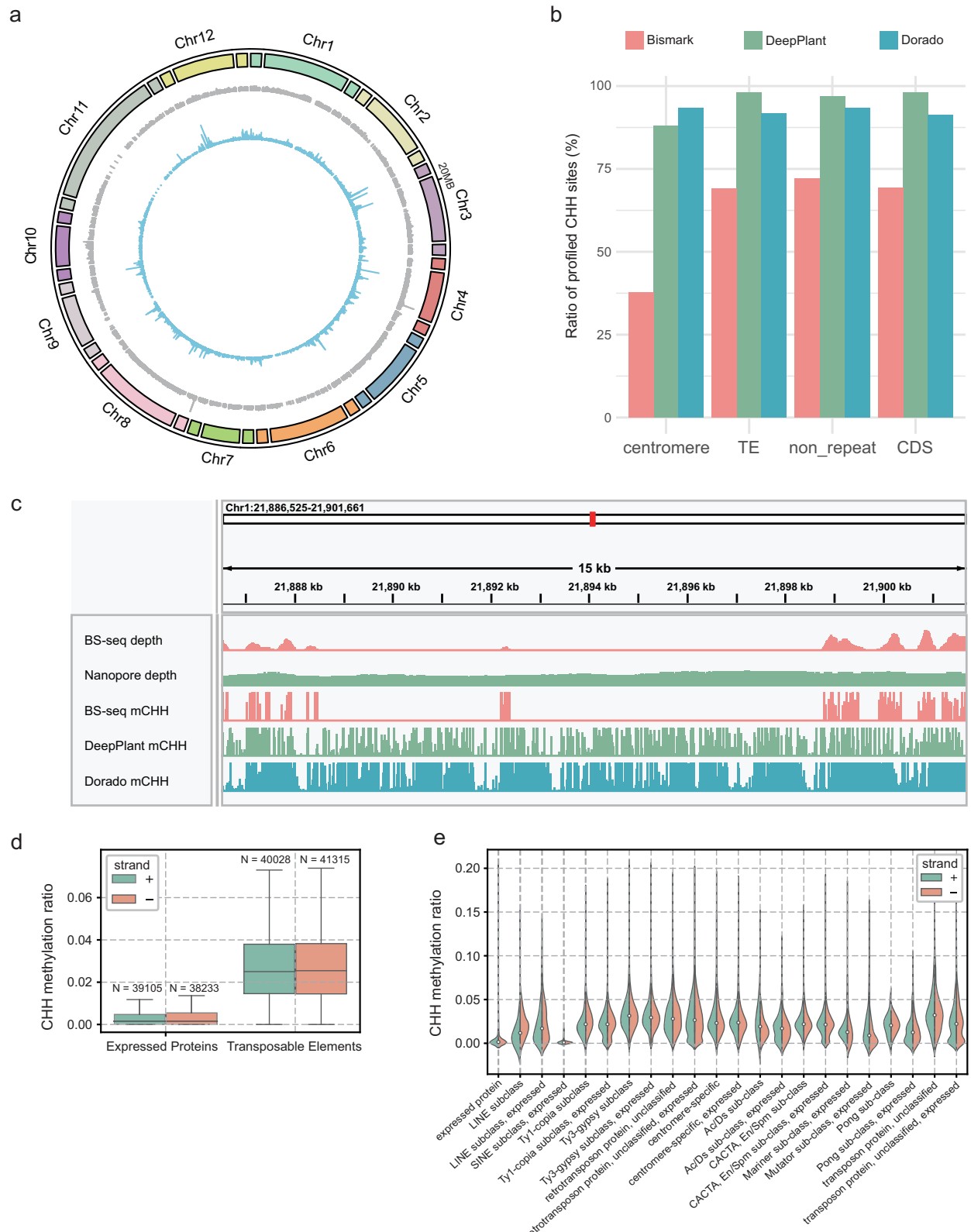

model training[42]. However, CHH methylation depends on RNA-directed DNA methylation (RdDM), which requires non-coding RNA guidance and the involvement of methyltransferases (DRM1 and DRM2), rendering in vitro enzyme-catalyzed DNA methylation ineffective[43]. PCR amplification with modified base substitutions[44] is a second choice; nevertheless, previous studies have reported significant difficulties replacing cytosines with high-purity 5mC in PCR

amplification[10,45]. And even if this approach succeeded the base contexts will be significantly different from native DNAs. In this study, we addressed the challenge by systematically analyzing publicly available BS-seq datasets and identifying plant species with abundant high-methylation CHH sites, such as *S. miltiorrhiza*, *S. tuberosum*, and *R. communis*. By generating new ONT R10.4 sequencing data for these species, we significantly increased the diversity and number of CHH-

**Fig. 3 | CHH methylation profiling of *O. sativa* centromere and transposable element regions. a** Circos plot illustrating DeepPlant CHH methylation profiling in centromeric regions and 100 kb intervals upstream and downstream in *O. sativa*. From outer to inner: ideograms of centromere (center box) and neighboring regions (two terminal boxes); histograms of normalized sequencing coverage across 100 bp bins (gray, normalized against mean genomic coverage); histograms of CHH methylation frequencies (blue) across 100 bp bins. **b** Comparison of CHH motif coverage ratios across different genomic regions between BS-seq, DeepPlant, and Dorado profiling. To be noticed, the same nanopore dataset was used for DeepPlant and Dorado profiling, and the coverage difference between DeepPlant and Dorado derived from the distinct read filters they applied. DeepPlant applies three thresholds for screening high-quality alignments, including MAPQ ≥ 20, primary alignment length/read length ≥80%, and mapping identity ≥80% by default. Only CHH motifs with a minimum read coverage of 10 were regarded as quantified. TE transposable element, CDS non-TE protein-coding sequences. **c** Read coverage and CHH methylation frequencies in the centromeric regions of Chr1, comparing whole-genome BS-seq data, DeepPlant, and Dorado analysis on Nanopore data. **d** Boxplot illustrating CHH methylation frequencies on the forward (+) and reverse (−) strands in protein-coding and transposable element (TE) regions. The center line represents the median; each box shows the first and third quartiles; minima represents the larger between Q1−1.5×IQR and the minimum observed value; maxima represents the smaller between Q3+1.5×IQR and the maximum observed value. **e** Violin plot displaying strand-specific CHH methylation status across various TE types and non-TE protein-coding regions. The annotation of TEs and protein-coding regions was acquired from T2T-NIP[40]. Source data of (**c**, **d**) are provided in Zenodo [https://doi.org/10.5281/zenodo.15062213]. Source data of the other panels are provided as a Source Data file.

positive samples. Our comprehensive training dataset now covers 97.2% of all possible 9-mer CHH contexts, averaging over 9225 samples per context−substantially surpassing DeepSignal-Plant which used *A. thaliana* and *O. sativa*[16]. This extensive coverage is crucial for training a model capable of generalizing across diverse plant species and methylation patterns.

In model training, by optimizing the model with 13-mer sequences, we achieved a balance between capturing sufficient sequence features and avoiding overfitting, which can be a risk when positive samples are limited. Importantly, DeepPlant demonstrates superior performance not only in CHH methylation detection but also in the CpG and CHG contexts. This consistent improvement across all contexts highlights DeepPlant's versatility and effectiveness in comprehensive 5mC detection in plants.

DeepPlant's enhanced performance extends to regions of the genome that are challenging for traditional BS-seq methods due to their repetitive nature and sequence complexity. For instance, we successfully profiled methylation patterns in most centromeric regions and TEs of *O. sativa*, achieving greater coverage than BS-seq and revealing strand-specific methylation patterns consistent with previous observations in *A. thaliana*[40]. DeepPlant's ability to quantify methylation states in a strand-specific manner provides valuable insights into the mechanisms of epigenetic regulation and the functional significance of asymmetric methylation patterns. These findings have implications for understanding the role of DNA methylation in regulating gene expression, transposon silencing, and genome stability in plants.

Despite these advancements, several limitations remain. The computational benchmark showed that DeepPlant is less computationally efficient than Dorado as it took much longer to call methylation on the same dataset (Supplementary Data 10). The scarcity of high-methylation CHH samples, although partially mitigated in this study, continues to pose challenges for model training and generalization. Our reliance on species with naturally abundant CHH methylation may not capture the full diversity of methylation patterns across all plant species. In addition, there is a need to address the potential for overfitting to specific sequence patterns, which underscores the importance of careful model optimization and validation. Future work should explore methods to artificially enrich CHH methylation samples, possibly through targeted methylation or synthetic biology approaches, to further enhance model training. Integrating DeepPlant with other epigenetic and genomic data could provide a more holistic understanding of epigenetic regulation. Applying DeepPlant to study epigenetic responses to environmental stresses, developmental cues, or pathogen interactions holds promise for advancing plant biology and agricultural sciences.

In conclusion, DeepPlant represents a significant advancement in plant epigenetics research, providing a powerful tool for accurate and comprehensive 5mC detection using ONT sequencing data. By overcoming limitations in CHH methylation detection, DeepPlant opens new avenues for exploring the complex epigenetic landscapes of plants. Its ability to profile methylation in challenging genomic regions enhances our capacity to study genome regulation, stability, and adaptation, ultimately contributing to advancements in plant epigenetics.

## Methods

### Public BS-seq and reference genomes

The reference genomes for all species were downloaded from NCBI (Supplementary Data 1). We reviewed relevant literature to obtain BS-seq data for *A. thaliana*[16], *O. sativa*[16], *B. vulgaris*[22], *S. miltiorrhiza*[23], *S. tuberosum*[24], *R. communis*[25–27], *C. sinensis*[28], *G. hirsutum*[29], *S. lycopersicum*[30], and *P. patens*[31–35] as detailed in Supplementary Data 1.

### Preparation of plant materials

Plant materials from various species were prepared for sequencing. Callus cultures were established from undeveloped ovules of *C. sinensis* cv. 'Liucheng'[46] and leaf discs of *Vitis vinifera* var. 'Baiti'[47]. Fresh roots of wild *S. miltiorrhiza* were collected in March 2024 from Song County, Henan, China, and the epidermal tissue was carefully sliced into thin sections (∼0.1 mm thick). For *A. thaliana* and *O. sativa*, leaves were collected from one-month-old seedlings of *A. thaliana* (L.) Heynh. Columbia-0 (Col-0) and *O. sativa* L. ssp. *Japonica* cv. Nipponbare. For *B. vulgaris* L. var. *cicla* and *Glycine max*, leaves were collected from 50-day-old plants. For *R. communis*, embryos were separated from fresh seeds of wild plants collected in March 2024 from Maoming, Guangdong, China. Sporangium powders of *M. polymorpha* L. and *P. patens* L. were purchased from the market, which later found to have low purity with <5% mapping ratio of BS-seq reads to reference genomes. Outer pericarps of *S. lycopersicum* (cultivar DRK0568) were dissected for DNA extraction. A tuber from the *S. tuberosum* variety A9, with the epidermis removed, was cut into 0.5-cm cubes. After these preparations, all plant samples were immediately frozen in liquid nitrogen and stored at −80 °C until DNA extraction.

### DNA extraction and nanopore sequencing

DNA extraction was carried out using DNeasy Plant Kits (Qiagen, Hilden, Germany) for the plant samples. Sequencing was performed on the Oxford Nanopore Technologies (ONT) PromethION R10.4.1 platform by Grandomics company (Wuhan, China). The raw nanopore data was base-called using ONT official basecaller, Dorado[19] (version 0.8.0), with the hac model, and 5mC modifications were called with the model version "dna_r10.4.1_e8.2_400bps_hac@v5.0.0" (https://github.com/nanoporetech/dorado?tab=readme-ov-file#dna-models). BS-seq and DeepPlant do not distinguish between 5mC and 5hmC modifications, therefore 5mC was called by Dorado using a 5mC/5hmC combined mode. We then aligned the reads to the reference genome using minimap2[48].

## Whole-genome BS-seq

For methylation profile consistency, the same genomic DNA used for nanopore sequencing was also subjected to BS-seq. The DNAs were first fragmented to an average size of 200−350 bp, followed by end-repair of the short DNA fragments. After bisulfite conversion, PCR amplification was performed, and the sequencing data were generated by BGI Genomics Co. Ltd. The sequencing reads were then processed using the standard workflow of Bismark (v0.24.0)[49]. Bismark provided a methylation call for every cytosine detected in CpG, CHG, and CHH contexts. The methylation frequency of cytosines is calculated as the number of mapped reads predicted to be methylated divided by the total number of mapped reads. To be noticed, the Bisamark pipeline only counts unique best alignments where the next best alignment does not exist or is not as good.

## Select high-confidence sites from BS-seq analyses

We used whole-genome BS-seq analysis as reference for selecting training samples. Sites for extracting training samples were selected from the whole genomes of *S. tuberosum*, *S. miltiorrhiza*, and *R. communis*. To ensure the reliability of the sites, we required the BS-seq coverage at each site to be above 5×. We selected CHH sites with a methylation frequency of zero as negative sites. Different criteria were applied for positive sites based on the methylation levels of different CHH motifs. For the CAA motif, we selected sites with a methylation frequency above 0.95, and the number of such sites exceeded 2 million. For the CAC, CAT, CTA, CTT, and CTC motifs, we selected sites with a methylation frequency above 0.9, with more than 1 million such sites. For the CCA, CCT, and CCC motifs, we selected sites with a methylation frequency above 0.85. For CHG motifs, positive sites were selected from *S. miltiorrhiza* and *R. communis* with methylation frequencies above 0.98, while negative sites were chosen with a frequency of zero. For CpG motifs, we selected sites from the HG002 and *B. vulgaris* datasets with methylation frequencies greater than 0.99 as positive sites and those with frequencies below 0.02 as negative sites.

## DeepPlant framework

DeepPlant uses raw sequencing signals from the Nanopore R10.4 flowcell and a reference genome as input data. The raw sequencing signals should be saved in the pod5 file format. The Nanopore sequencing signals need to be base-called by a basecalling tool to obtain the corresponding read sequences, which are then aligned with the reference genome using an alignment tool. The reads need to be stored in a BAM file, and the move table must be retained during the basecalling to enable correspondence between the bases and their sequencing signals. After basecalling, the reads in the BAM file should be sorted by the pod5 file names (fn field).

## Feature extraction

We locate the cytosine sites in the aligned reads based on the selected reference genome's cytosine positions. A k-mer is extracted from the read for each target cytosine, with the target cytosine positioned in the middle of the k-mer. Reads not aligned to the reference or had low alignment quality were filtered out. The default filtering criteria for low-quality alignments are as follows: Reads with a mapping quality (MAPQ) score of less than 20 were filtered. Reads were further filtered if the length of the primary alignment (total length minus soft clipped bases) to the total number of bases in the read was less than 80% or the mapping identity was lower than 80%. We locate the raw sequencing signals for the k-mer from the pod5 file using the move table. The raw sequencing signals are standardized using the median shift and median absolute deviation (MAD) scale[36]. The mean and standard deviation are calculated for each base's standardized signal. These, along with basecalling quality, the number of corresponding signals, and the base itself, form the sequence features. This leads to a matrix with

dimensions of k×5. In addition, 15 signals are sampled from the standardized signals of each base to form signal features, resulting in a k×15 matrix. Thus, each k-mer has two types of features for cytosine methylation detection.

## Model architecture

The k-mer sequence is transformed into an embedding representation, combined with other statistical features to create the sequence features. We then use three encoders to build DeepPlant. The first encoder independently encodes the sequence features, while the second encoder independently encodes the signal features. The encoded results from both are concatenated and fed into the third encoder, which performs collaborative encoding of the sequence. After the collaborative encoding, a feedforward network is used as the final classifier to determine the methylation probability of the target cytosine.

DeepPlant encoders use two structures: a bidirectional recurrent neural network(BiRNN)[11] consisting of long short-term memory(LSTM)[12] units and a transformer encoder[13]. The bidirectional LSTM scans the k-mer both forward and backward. Then, a feedforward network produces hidden vectors, aggregating information from all bases in the k-mer at the end of the sequence. We extract the hidden vectors from the sequence's end in both the forward and backward directions and concatenate them to get the final encoding representation. On the other hand, when using the transformer encoder in DeepPlant, since the attention mechanism does not retain positional information, similar to natural language processing, both the sequence feature encoder and the signal feature encoder need to perform positional encoding at the beginning. We use sinusoidal positional encoding[13], which can be described as:

$$
\begin{cases}
PE_{(pos, 2i)} = \sin\left(pos \cdot 10000^{-2i/d}\right), \\
PE_{(pos, 2i+1)} = \cos\left(pos \cdot 10000^{-2i/d}\right)
\end{cases} \tag{1}
$$

where pos is the position of the base within the k-mer, $i$ is the index of the hidden vector dimension of the base or signal, and d is the dimension of the hidden vector with default value of 128.

Then, we adopt a structure similar to BERT[50], using a multi-head attention module and a feedforward network to construct another encoder with residual connections and layer normalization[51] between the modules. The transformer encoder attends to the influence of the neighboring cytosines on both sides of the base, which affects its signal. We extract the hidden vector as the final encoding representation at the central position.

## Training

We ultimately extracted 124 million samples genome-wide from *S. miltiorrhiza*, *S. tuberosum*, and *R. communis* nanopore sequencing data for CHH motifs, with a 1:1 ratio of positive to negative samples. For CHG motifs, we extracted samples from *S. miltiorrhiza* and *R. communis*, and for CpG motifs, samples were extracted from the HG002 and *B. vulgaris* datasets. About 1% of the total samples were used as a test set to select the best-performing model. Adam[52] was used as the optimizer for the network, with exponential decay rates for the first and second-moment estimates set to 0.9 and 0.999, respectively. The initial learning rates for the LSTM and transformer encoder were set to 0.001 and 0.0005, respectively, and decreased by 80% with each epoch. The model's optimization gradients were generated using cross-entropy loss. Gradient clipping was applied to prevent gradient explosion in the network. In addition, dropout layers[53] were inserted between different layers of the model to mitigate overfitting, and early stopping[54] was employed during training. The sequence feature encoder and the signal feature encoder were set to 2 layers, while the collaborative encoder was set to 3. The sequence encoder and signal

encoder had a feature dimension of 128, whereas the collaborative encoder had a feature dimension of 256. Detailed network parameters are listed in Supplementary Data 4.

## Model evaluation

We evaluated DeepPlant and Dorado using Nanopore R10.4 sequencing data from nine species: *B. vulgaris, O. sativa, A. thaliana, V. vinifera, C. sinensis, S. miltiorrhiza, S. tuberosum, S. lycopersicum,* and *R. communis.* We conducted all evaluations for each tool independently, and each tool was used with its default parameter settings. For read-level evaluation, we used BS-seq analysis as the benchmark. We selected sites with sequencing coverage higher than 5×, where sites with a methylation frequency of 0 were used as negative samples and those with a methylation frequency of 100% as positive samples. We extracted k-mer samples from these sites, sampling 100,000 positive and 100,000 negative samples. For datasets with insufficient positive samples, we applied the Synthetic Minority Over-Sampling Technique (SMOTE)[55] to oversample and meet the evaluation requirements. We used DeepPlant and Dorado to determine their methylation status, and the sample is classified as methylated if the methylation probability is higher than the non-methylation probability. To increase the reliability of the results, we repeated this process three times and calculated the average of the three evaluation results.

For site-level evaluation, we downsampled the nanopore sequencing data to obtain datasets with depth of 5×, 10×, 15×, 20×, 25×, and 30×. For each sequencing depth, we selected cytosine locus with BS-seq and nanopore sequencing coverage, both higher than 5×, as valid evaluation sites, the number of sites detected by each method is listed in Supplementary Data 2. We determine the methylation threshold $P_{th}$ based on the output methylation probability distribution (Supplementary Fig. 2). Divide the range from 0.2 to 0.9 into 70 intervals with a step size of 0.01, using the left endpoint of each interval as the representative value for that interval. For the detection results of a single dataset, group the samples into the corresponding intervals based on their methylation probabilities, count the number of samples in each interval and calculate their proportion. The value corresponding to the interval with the smallest proportion is selected as the methylation threshold $P_{th}$. If the methylation probability $P_m \geq P_{th}$, the sample is classified as methylated; otherwise, it is classified as non-methylated. After aligning the target cytosine in the test reads with the reference genome, we calculated the number of cytosines predicted to be methylated and the total number of cytosines at each target genomic site to determine the methylation frequency at the site. We then calculate the Pearson correlation between the predicted methylation frequency at the whole-genome evaluation sites and the methylation frequency from BS-seq. The benchmarking results, including runtime and memory consumption, are provided in Supplementary Data 10.

## K-mer balancing

Due to the low methylation levels of the CHH motif, there is a significant imbalance between the number of positive and negative k-mer samples available for training. This causes the model to produce different prediction biases for different k-mers, leading to unstable performance. Compared to DeepSignal-plant[16], we adopted a stricter sample balancing method to mitigate the impact caused by k-mer sequences. The algorithm is as follows:

Input: a set of positive samples $S_{pos}$, set of negative samples $S_{neg}$, the maximum quantity of kmer $k\_max$.

Output: a set of balanced positive samples $S'_{pos}$, set of balanced negative samples $S'_{neg}$.

1. $K_{pos}$ = set of k-mers in $S_{pos}$, $K_{neg}$ = set of k-mers in $S_{neg}$
2. $K_{comm} = K_{neg}$. Intersection($K_{pos}$)
3. $KNUM_{pos}$ = number of samples of each k-mer in $S_{pos}$, $KNUM_{neg}$ = number of samples of each k-mer in $S_{neg}$
4. $S'_{neg} = \varnothing$, $S'_{pos} = \varnothing$

5. for each k-mer k in $K_{comm}$ do
   (1) $k\_count$ = min of $KNUM_{pos}(k)$, $KNUM_{neg}(k)$, $k\_max$
   (2) $S'_{pos\_k}$ = set of at most $k\_count$ samples of k extracted from $S_{pos}$ randomly
   (3) $S'_{neg\_k}$ = set of at most $k\_count$ samples of k extracted from $S_{neg}$ randomly
   (4) $S'_{neg}$ += $S'_{neg\_k}$
   (5) $S'_{pos}$ += $S'_{pos\_k}$
6. return $S'_{pos}$, $S'_{neg}$

## Data availability

All sequencing data generated in this study (BS-seq and nanopore sequencing data of *S. miltiorrhiza, S. tuberosum, R. communis, C. sinensis, S. lycopersicum,* and *V. vinifera;* BS-seq data of *G. max, P. patens* and *M. polymorpha*) and our assembly of *V. vinifera* have been deposited in the Genome Sequence Archive in BIG Data Center, Beijing Institute of Genomics under accession PRJCA030666. The BS-seq and Nanopore sequencing data of *A. thaliana, O. sativa,* and *B. vulgaris* are available at BIG under accession PRJCA023349. The reference genomes of *S. miltiorrhiza* (GCF_028751815.1), *S. tuberosum* (GCF_000226075.1), *R. communis* (GCF_019578655.1), *C. sinensis* (GCF_022201045.2), *A. thaliana* (GCF_000001735.4), *O. sativa* (GCF_034140825.1), *S. lycopersicum* (GCA_915070445.1), and *B. vulgaris* (GCF_026745355.1) were downloaded from NCBI. The genome assembly and annotation for the T2T-NIP of *O. sativa* were accessed from RiceSuperPIRdb [http://www.ricesuperpir.com/web/nip]. Source data for Fig. 3c and d as well as Supplementary Fig. 3 and 6 are provided in Zenodo [https://doi.org/10.5281/zenodo.15062213]. Source data are provided with this paper.

## Code availability

DeepPlant codes, installation, and usage instructions are available at Github [https://github.com/xiaochuanle/DeepPlant] and Zenodo [https://doi.org/10.5281/zenodo.15022822], which are distributed under the MIT License.

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

## Acknowledgements

The authors thank all those who generated and freely released the data analyzed in our present study. The authors acknowledge financial support from the National Key R&D Program of China (2022YFF1201900 to C.-L.X.), the National Natural Science Foundation of China (no. U21A20414 to Q.H.; no. 32270713, 62350004 to C.-L.X.; no. 82230031 to W.C.); Guangdong Basic and Applied Basic Research Foundation (2020B1515020057 to C.-L.X.).

## Author contributions

C.-L.X., L.C., Q.H., and W.C. conceived the study. H.-X.C., C.-L.X., and Z.-D.L. implemented the algorithms of DeepPlant, H.-X.C., and X.B. wrote the code of DeepPlant, X.B., H.-X.C., B.W., R.S., and H.-C.Y. carried out experiments and data analysis, H.-X.C., B.W., X.B., C.-L.X., and Y.C. wrote the manuscript. B.W. modified and improved the manuscript. All authors read and approved the final version of the manuscript.

## Competing interests

The authors declare no competing interests.
