## [Peer Review file · Nature Communications]

Accurate Cross-Species 5mC Detection for Oxford Nanopore Sequencing in Plants with DeepPlant

Corresponding Author: Dr Chuan-Le Xiao

Version 0:

Reviewer comments:

Reviewer #1

(Remarks to the Author)

In the paper "DeepPlant: An Accurate Cross-Species 5mC Detection Tool for Oxford Nanopore Sequencing in Plants," He-Xu Chen et al. present a new approach for improving 5-methylcytosine (5mC) detection in plant genomes. The key contributions include generating and publishing new plant datasets using ONT R10.4.1 and WGBS, which are valuable for training models, as well as providing pre-trained models for detecting 5mC in CpG, CHG, and CHH contexts. While the model architecture is similar to existing tools like DeepSignal, the strength of this work lies in the inclusion of diverse datasets and the tool specifically designed for plant epigenomics and the ONT R10.4.1 pore type. The authors compare DeepPlant to the Dorado model, which also detects 5mC in CpG, CHG, and CHH motifs, showing that DeepPlant outperforms Dorado on most metrics, particularly in the CHH context, which is unique to plant genomes. Overall, the paper is well-written and clear, making it easy to follow and understand.

Concerns:

A major concern with the evaluation presented in this paper is the inclusion of training data in the evaluation tables. For example, the reported Pearson's correlation values between DeepPlant predictions and BS-seq data at 30× sequencing depth range from 0.705 to 0.881 across all evaluated datasets. However, the highest correlation value (0.881) is achieved for a species that was included in the training set (*R. communis*). This undermines the validity of the evaluation, as models often perform significantly better on data they were trained on compared to unseen datasets. The inclusion of training data in the evaluation skews the results, making it difficult to assess the model's true generalization capability across new species. A clear separation between training and evaluation datasets is essential for a reliable assessment of model performance.

A concern with the current model design is the lack of systematic ablation analysis to evaluate the contributions of its components. The model is composed of three main parts: the first encodes sequence features (including signal length, base qualities, mean, standard deviation, and the base at each position in a k-mer), the second encodes signal features (normalized signal), and the third combines and processes the outputs from the first two parts. While the architecture is well-designed and incorporates diverse feature representations, it's unclear how much each feature and encoder contributes to the final predictions. Performing a systematic ablation study—by removing specific features or encoders—would provide crucial insights into the importance of each component, highlight potential redundancies, and help refine the architecture for improved performance.

A significant concern is the omission of *Glycine max* (*G. max*) from the analysis without any explanation. The authors state: "We then collected tissue samples from six of these species (excluding *S. lycopersicum* due to its close relation to *S. tuberosum*) and conducted BS-seq. For better sample diversity, *A. thaliana*, *O. sativa*, *Glycine max*, *Vitis vinifera*, and *Marchantia polymorpha* were also added to the analysis." However, while *S. miltiorrhiza*, *S. tuberosum*, and *R. communis* were used for training, and five additional species (*A. thaliana*, *V. vinifera*, *O. sativa*, *B. vulgaris*, and *C. sinensis*) were included for evaluation, *G. max* is not mentioned in the final analysis, nor is its exclusion explained.

In contrast, the authors explicitly state that *P. patens* and *M. polymorpha* were excluded due to possible sample impurities, which is reasonable. However, Figure 1 includes ratios of high-methylation CHH sites for all excluded datasets, including these two species, indicating that they were used in the manuscript. Given their inclusion in the analysis, it would be appropriate for the authors to publish these datasets, regardless of their exclusion from training and evaluation, to support

transparency and reproducibility. Additionally, since G. max was part of the original sample collection, its data should either be included in the analysis or its omission clearly justified.

For the read-level evaluation, the authors subsample 100,000 positive and 100,000 negative examples to create a balanced dataset, ensuring an equal number of positive and negative samples. For datasets where there are not enough positive examples to reach this number, they apply SMOTE (Synthetic Minority Over-sampling Technique) to artificially generate additional positive samples. While this approach helps create a balanced dataset for evaluation, it alters the natural distribution of the data, potentially making the evaluation less reflective of real-world conditions, where such balance rarely exists. This distribution shift could bias the results, as the model might perform differently on balanced datasets compared to naturally imbalanced data. To address this, it would be beneficial to include results on the full, unsampled datasets to provide a more realistic and unbiased evaluation of the model's true performance on real-world data.

The authors state, "To increase the reliability of the results, we repeated this process five times and calculated the average of the five evaluation results." However, they do not report the standard deviation or any other measure of variability to accompany the mean.

The authors do not specify whether the evaluations were performed on examples successfully called by both Dorado and DeepPlant (i.e., the intersection of valid predictions for both tools) or if the evaluations were conducted independently for each tool. Regardless, the number of evaluated examples should be explicitly reported for clarity and transparency.

The method for determining methylation status in a site-level evaluation is unclear. The authors state: "We determine the methylation threshold P_{th} based on the output methylation probability distribution, selecting the probability value within the range of 0.2 to 0.9 with the minor proportion in the probability distribution as P_{th} . If the methylation probability $P_m \geq P_{th}$, the sample is classified as methylated; otherwise, it is classified as non-methylated." However, it is not explained how the threshold is chosen in practical terms, particularly how the "minor proportion" is defined or calculated within the specified range. Greater clarity on the selection process is needed to fully understand this approach.

Only Dorado is used for comparison in the detection of 5mC in CpG motifs, despite the availability of other tools like DeepMod and Rockfish. While the authors exclude older tools such as Tombo, Megalodon, and DeepSignal-Plant due to their incompatibility with R10.4, this reasoning does not apply to DeepMod and Rockfish. The authors should explain why Dorado was chosen as the sole comparison tool or include comparisons with these additional methods to provide a more thorough evaluation of DeepPlant's performance in CpG detection.

The paper does not provide information about the running time or memory requirements of DeepPlant. While performance metrics like accuracy and F1 score are essential, practical factors such as computational efficiency and resource usage are equally important. Reporting details on running time and memory consumption would offer valuable insights into the model's scalability and suitability for large-scale or resource-constrained applications.

In Figure 3B, it is unclear whether the ratio of profiled sites corresponds to Nanopore coverage or DeepPlant coverage. The authors state that "Reads not aligned to the reference or had low alignment quality were filtered out." From this, it can be inferred that Nanopore coverage is likely higher than DeepPlant coverage. Specifying the coverage obtained for Dorado would also be helpful. Furthermore, the quality threshold for filtering low-quality alignments is not explicitly defined in the manuscript. However, in the code (GitHub commit: 67db8ef; Call_Modification.cpp, lines 107–109), the authors define three constants for filtering, one of which (`mapq_thresh_hold`) appears to set the mapping quality threshold at 20. More detailed information about the other two heuristics used for filtering is necessary to fully understand the criteria applied during the analysis.

(Remarks on code availability)

The installation process described on GitHub is problematic and was unsuccessful. Despite having CUDA 11.8 installed and correctly configured (with `'nvcc -version'`, `'torch.version.cuda'`, and `'torch.cuda.is_available()'` returning the expected output), an error occurs when running `cmake`. The first line of the error reads: "Could NOT find CUDA (missing: CUDA_INCLUDE_DIRS CUDA_CUDART_LIBRARY) (found version '11.8')." The installation was performed on a server with the `'cuda/11.8.0'` and `'cudnn/8.9.2_cu1'` module loaded.

Additionally, the installation instructions for CUDA are not user-friendly, as they assume the user has root privileges, which may not always be the case. It's also worth noting that running `'pip install torch==2.0.1'` without the `'--index-url https://download.pytorch.org/whl/cu118'` flag installs PyTorch for CUDA version 11.7. Providing a Dockerfile, a simpler execution option, or clearer setup instructions would improve accessibility and ensure a smoother installation process.

Reviewer #2

(Remarks to the Author)

(Remarks on code availability)

The installation process on GitHub is problematic and fails despite CUDA 11.8 being correctly configured. Errors occur during cmake, and the instructions assume root privileges, which may not be available to all users. Additionally, installing PyTorch without specific flags defaults to CUDA 11.7, causing potential compatibility issues. Providing a Dockerfile, a simpler installation process, or clearer instructions would greatly improve accessibility and reliability.

Reviewer #3

(Remarks to the Author)

Major Revisiong:

The authors developed a deep learning model to process DNA methylome data generated from ONT R10.4 platform. Particularly, they used mCHH enriched genome in the training to improve the accuracy of mCHH data detection. In general, the new model provides valuable tool for the application of ONT R10.4 in plant epigenome sequencing. I have several concerns:

1)When the authors processed the BS-seq data, how did they treat with the multiple mapped reads? These reads could be generated from repeat enriched regions, such as centromere. if they discard all multiple mapped reads, it will lead to underestimation of mCHH caption in BS-seq data., such as in figure 3b and 3c.

2)In figure 3B and 3C, besides BS-seq and Deepplant/Nanopore, the Nanopole data analysis based on Dorodo should also be included to see if there is any improvement using Deepplant then using Dorado.

3)All the analysis are statistical results. The authors should show several screenshots of BS-seq, Dorado, Deepplant data in regions where contain mCG, mCHG and mCHH, and the authors could show these three contexts respectively, to demonstrate that the capture of mCHH is more accurate in Deepplant processed dataset.

(Remarks on code availability)

Version 1:

Reviewer comments:

Reviewer #1

(Remarks to the Author)

We appreciate the authors for taking the time to incorporate our suggestions and improve the clarity and completeness of their manuscript. The clear separation of training and evaluation datasets, along with the additional explanations regarding dataset exclusions and public availability, significantly enhance the transparency of the study. The ablation study provides valuable insight into the contribution of individual components, and the inclusion of variability measures strengthens the reliability of the reported results. We also appreciate the clarification on the exclusion of DeepMod2 and the detailed information on DeepPlant's runtime and memory requirements. These thoughtful revisions contribute to a more comprehensive and well-supported analysis.

Concern 1:

A remaining major concern is balanced sampling for read-level evaluation. For evaluation to be truly informative, results should be presented on real-world, naturally imbalanced datasets. Artificially balancing datasets during evaluation does not help users assess a method's practical applicability, as it distorts the class distribution and can inflate performance metrics. While we agree that some metrics, such as accuracy and AUC-ROC, are not suitable for evaluating highly imbalanced datasets, we cannot agree with the claim that metrics like AP, F1-score, precision, and recall "fail to accurately reflect true performance." These metrics are widely used in the machine learning and bioinformatics communities for precisely this purpose—evaluating models in extreme class imbalance scenarios.

Precision, in particular, can be "unforgiving" when there is a significant imbalance, but this does not make it unrepresentative or biased. Instead, low precision highlights a model's struggle to distinguish true positives from false positives. This is valuable information for users who care about the reliability of positive predictions. Moreover, metrics like average precision (AP) and precision-recall (PR) curves provide an even clearer picture, as they evaluate how well a model ranks positive examples rather than just how it labels them.

In summary, artificially balancing datasets for evaluation risks misleading users about real-world performance. Instead, presenting results on naturally imbalanced datasets—using well-established metrics like precision, recall, F1-score, and AP—ensures transparency and allows users to make informed decisions about method suitability.

#Concern 2:

We appreciate the authors for providing a detailed description of their method for determining methylation status in a site-level evaluation. While the procedural explanation is now clearer, the specific application of this method remains ambiguous. Is it applied to WGBS data, DeepPlant calls, or both?

Additionally, if this method is applied to DeepPlant, does the term methylation probability refer to the site-level prediction (i.e., the number of methylated Cs divided by the total number of Cs at that site), or does it correspond directly to the model's output? If the threshold is determined using WGBS data, is the same threshold applied to DeepPlant predictions? Furthermore, if this method is applied to DeepPlant but not Dorado, it would be valuable to evaluate its impact on model performance by applying it to Dorado or by comparing it with a simpler thresholding approach for DeepPlant predictions.

Minor comments:

We thank the authors for clearly separate training and evaluation datasets. However, some figures could still benefit from having clear separation, e.g. Supplementary Fig. 3

(Remarks on code availability)

No additional remarks

Reviewer #2

(Remarks to the Author)

(Remarks on code availability)

The instructions for running the tool using Docker are clearly written. However, the run command is missing the repository prefix (image name should be given with 'chenhx26/deeplint:1.1.0' instead of just 'deeplint:1.1.0'), which may cause an error when users attempt to run the image.

Reviewer #3

(Remarks to the Author)

After reviewing the manuscript, I am pleased to suggest that that I have no revisions or objections to raise. The content is well-presented, and it contributes valuable insights to the field. Therefore, I recommend proceeding with its acceptance. Thank you for considering my feedback. Should you need any further information or discussion, please do not hesitate to contact me.

(Remarks on code availability)

Accept

Version 2:

Reviewer comments:

Reviewer #1

(Remarks to the Author)

We acknowledge the authors' efforts in incorporating the suggestions and improving the clarity of the manuscript. The addition of evaluations on datasets that were not sampled is particularly valuable. Furthermore, their explanation of the method for determining methylation status in site-level evaluations is satisfactory. We have no further concerns regarding the manuscript.

(Remarks on code availability)

We included this in previous reviews.

Reviewer #2

(Remarks to the Author)

(Remarks on code availability)

Reviewer #1 (Remarks to the Author):

In the paper "DeepPlant: An Accurate Cross-Species 5mC Detection Tool for Oxford Nanopore Sequencing in Plants," He-Xu Chen et al. present a new approach for improving 5-methylcytosine (5mC) detection in plant genomes. The key contributions include generating and publishing new plant datasets using ONT R10.4.1 and WGBS, which are valuable for training models, as well as providing pre-trained models for detecting 5mC in CpG, CHG, and CHH contexts. While the model architecture is similar to existing tools like DeepSignal, the strength of this work lies in the inclusion of diverse datasets and the tool specifically designed for plant epigenomics and the ONT R10.4.1 pore type. The authors compare DeepPlant to the Dorado model, which also detects 5mC in CpG, CHG, and CHH motifs, showing that DeepPlant outperforms Dorado on most metrics, particularly in the CHH context, which is unique to plant genomes. Overall, the paper is well-written and clear, making it easy to follow and understand.

Concerns:

A major concern with the evaluation presented in this paper is the inclusion of training data in the evaluation tables. For example, the reported Pearson's correlation values between DeepPlant predictions and BS-seq data at 30× sequencing depth range from 0.705 to 0.881 across all evaluated datasets. However, the highest correlation value (0.881) is achieved for a species that was included in the training set (*R. communis*). This undermines the validity of the evaluation, as models often perform significantly better on data they were trained on compared to unseen datasets. The inclusion of training data in the evaluation skews the results, making it difficult to assess the model's true generalization capability across new species. A clear separation between training and evaluation datasets is essential for a reliable assessment of model performance.

Response:

Thank you for the valuable suggestions. We have now clearly separated the training and evaluation datasets in this revision. For fairness, when describing the model evaluation results, we have now excluded the three datasets used for training in the abstract and the main manuscript. To better illustrate the model's generalization capability, we sequenced new BS-seq and nanopore dataset for *S. lycopersicum* and further compared DeepPlant and the previous state-of-art software Dorado on it. Still, the results support that DeepPlant has advantage on the CHH motif. To be noticed, the correlation coefficients are affected by many factors such as mapping rate, reference genome divergence level, and methylation frequency distribution, making direct comparison of correlation coefficients across datasets less meaningful. After all, the evaluation results showed that the DeepPlant showed even greater advantage compared to Dorado on several test-only datasets than on the training datasets, suggesting better generalization capability across tested datasets. We hope these revisions (in lines 42-43, 108, and et al.) have made the evaluation conclusion more robust.

A concern with the current model design is the lack of systematic ablation analysis to evaluate the contributions of its components. The model is composed of three main parts: the first encodes sequence features (including signal length, base qualities, mean, standard deviation, and the base at each position in a k-mer), the second encodes signal features (normalized signal), and the third combines and processes the outputs from the first two parts. While the architecture is well-designed and incorporates diverse feature representations, it's unclear how much each feature and encoder contributes to the final predictions. Performing a systematic ablation study—by removing specific features or encoders—would provide crucial insights into the importance of each component, highlight potential redundancies, and help refine the architecture for improved performance.

Response:

Thank you for the suggestions. Understanding the importance of each component indeed will deepen the insights of this study. Accordingly, we have conducted systematic ablation experiments on DeepPlant to evaluate the contribution of its triple-encoder. By removing each of the encoders and training the model using the same samples and conditions, we quantitatively assessed the resulted models on the *O. sativa* and *A. thaliana* datasets. The results, presented in Supplementary Table 3, demonstrated that all three encoders have played a critical role in the model's performance. Among them, the sequence encoder has the greatest impact on the results, while the combine encoder has the least. For instance, the removal of any encoder resulted in at least a 15% decrease in performance on the *A. thaliana* dataset, indicating that each encoder is indispensable. We have revised the manuscript in lines 171-173.

A significant concern is the omission of *Glycine max* (*G. max*) from the analysis without any explanation. The authors state: "We then collected tissue samples from six of these species (excluding *S. lycopersicum* due to its close relation to *S. tuberosum*) and conducted BS-seq. For better sample diversity, *A. thaliana*, *O. sativa*, *Glycine max*, *Vitis vinifera*, and *Marchantia polymorpha* were also added to the analysis." However, while *S. miltiorrhiza*, *S. tuberosum*, and *R. communis* were used for training, and five additional species (*A. thaliana*, *V. vinifera*, *O. sativa*, *B. vulgaris*, and *C. sinensis*) were included for evaluation, *G. max* is not mentioned in the final analysis, nor is its exclusion explained.

Response:

Thank you pointing it out. We initially sequenced BS-seq and Pacbio Hifi (for another project) for the *G.max* var. Huachun No.6, which showed high heterozygosity and abundant nucleotide variances to the reference genome (GCF_000004515.6). The high difference level compared to the reference will affect the quality of BS-seq that uses base conversion for detection of methylation state, therefore we have not sequenced nanopore data for it. We have now explained it in the manuscript (in lines 148-149) and uploaded the BS-seq data to public database as requested (Data availability statement updated).

In contrast, the authors explicitly state that *P. patens* and *M. polymorpha* were excluded due to possible sample impurities, which is reasonable. However, Figure 1 includes ratios of high-methylation CHH sites for all excluded datasets, including these two species, indicating that they were used in the manuscript. Given their inclusion in the analysis, it would be appropriate for the authors to publish these datasets, regardless of their exclusion from training and evaluation, to support transparency and reproducibility. Additionally, since *G. max* was part of the original sample collection, its data should either be included in the analysis or its omission clearly justified.

Response:

Thank you for the comments. We have uploaded our BS-seq data for *P. patens* and *M. polymorpha*, which are available at <http://gsa.big.ac.cn> under Project Accession No. PRJCA030666. As mentioned above, we have also uploaded the BS-seq data of *G. max*.

For the read-level evaluation, the authors subsample 100,000 positive and 100,000 negative examples to create a balanced dataset, ensuring an equal number of positive and negative samples. For datasets where there are not enough positive examples to reach this number, they apply SMOTE (Synthetic Minority Over-sampling Technique) to artificially generate additional positive samples. While this approach helps create a balanced dataset for evaluation, it alters the natural distribution of the data, potentially making the evaluation less reflective of real-world conditions, where such balance rarely exists. This distribution shift could bias the results, as the model might perform differently on balanced datasets compared to naturally imbalanced data. To address this, it would be beneficial to include results on the full, unsampled datasets to provide a more realistic and unbiased evaluation of the model's true performance on real-world data.

Response:

Thank you for the comments. Following your suggestions, we evaluated the single-molecule performance of DeepPlant and Dorado on three full unbalanced datasets: *A. thaliana* (positive sample ratio 0.0506%), *B. vulgaris* (0.1249%), and *O. sativa* (0.033%). The results are shown in the **table below**. Due to the extreme scarcity of positive samples, the evaluation metrics AP, F1, Precision, and Accuracy fail to accurately reflect the true performance of the models. This is because even if only a very small proportion of negative samples are misclassified as positive (false positives), their number would significantly exceed the number of true positive samples, leading to a decline in both F1 and Precision metrics. Similarly, the AP value, which is calculated based on the precision-recall curve, is also reduced as a result. Since the vast majority of samples are negative, Accuracy shows a substantial increase. In fact, under such circumstances, Accuracy largely reflects the model's ability to classify negative samples correctly. When the model correctly classifies all negative samples, its Accuracy can exceed 99.8%, rendering this metric unrepresentative of the model's overall performance. For the reasons, using extreme unbalanced datasets for single-molecule evaluations reduces is also severely biased. Therefore, we prefer to use balanced datasets to reflect the

single-molecule performance of the models. Additionally, to reflect the real-world sample distribution, we used the full datasets for site-level (methylation frequency) evaluation, which provides some reference value for practical scenarios.

	tools	AUC	AP	F1-score	Precision	Accuracy	Recall	Specificity
A. thaliana	DeepPlant	0.8668	0.0073	0.001373	0.000687	0.97023	0.669626	0.970239
	Dorado	0.8492	0.0024	0.001231	0.000616	0.969593	0.625466	0.969604
B. vulgaris	DeepPlant	0.933079	0.056785	0.007849	0.003943	0.931204	0.835183	0.931235
	Dorado	0.880859	0.010706	0.004315	0.002164	0.901871	0.733399	0.90192
O. sativa	DeepPlant	0.9191	0.1076	0.026955	0.013706	0.947117	0.808072	0.947243
	Dorado	0.907942	0.01916	0.008923	0.004486	0.877289	0.813208	0.877333

The authors state, "To increase the reliability of the results, we repeated this process five times and calculated the average of the five evaluation results." However, they do not report the standard deviation or any other measure of variability to accompany the mean.

Response:

Thank you for pointing it out. We noticed the lack of rigor in this regard; therefore, we calculated the standard deviation of all single-molecule evaluation results, as shown in revised Supplementary Tables 6 and 7.

The authors do not specify whether the evaluations were performed on examples successfully called by both Dorado and DeepPlant (i.e., the intersection of valid predictions for both tools) or if the evaluations were conducted independently for each tool. Regardless, the number of evaluated examples should be explicitly reported for clarity and transparency.

Response:

Thank you for pointing it out. We conducted all evaluations for each tool independently. For single-molecule evaluations, we extracted 200,000 samples under the same criteria and repeated the process five times, which has now been described in lines 498-505. For quantitative evaluations, we additionally provided the number of sites detected for three types of motifs by each method, as shown in Supplementary Table 2.

The method for determining methylation status in a site-level evaluation is unclear. The authors state: "We determine the methylation threshold P_{th} based on the output methylation probability distribution, selecting the probability value within the range of 0.2 to 0.9 with the minor proportion in the probability distribution as P_{th} . If the methylation probability $P_m \geq P_{th}$, the sample is classified as methylated; otherwise, it is classified as non-methylated." However, it is not explained how the threshold is chosen in practical

terms, particularly how the "minor proportion" is defined or calculated within the specified range. Greater clarity on the selection process is needed to fully understand this approach.

Response:

Thank you. We have described the detailed procedure in the Methods section now. The method applied is as follows in lines 510-518:

'Divide the range from 0.2 to 0.9 into 70 intervals with a step size of 0.01, using the left endpoint of each interval as the representative value for that interval. For the detection results of a single dataset, group the samples into the corresponding intervals based on their methylation probabilities, count the number of samples in each interval, and calculate their proportion. The value corresponding to the interval with the smallest proportion is selected as the methylation threshold.'

Only Dorado is used for comparison in the detection of 5mC in CpG motifs, despite the availability of other tools like DeepMod and Rockfish. While the authors exclude older tools such as Tombo, Megalodon, and DeepSignal-Plant due to their incompatibility with R10.4, this reasoning does not apply to DeepMod and Rockfish. The authors should explain why Dorado was chosen as the sole comparison tool or include comparisons with these additional methods to provide a more thorough evaluation of DeepPlant's performance in CpG detection.

Response:

Thank you for the suggestions. Accordingly, we further evaluated the recently published in this revision. For DeepMode2, because its original study (*Ahsan et al.*,) has carried out multiple assessments showing that it performed not as well as Dorado on R10.4 nanopore data, we have not included it in comparison in this revision. The results have been provided in Supplementary Table 6 and still identify DeepPlant as one of the state-of-art tools on CpG detection. The new benchmark results have also been updated in the manuscript.

Ahsan, M.U., Gouru, A., Chan, J. et al. A signal processing and deep learning framework for methylation detection using Oxford Nanopore sequencing. Nat Commun 15, 1448 (2024). <https://doi.org/10.1038/s41467-024-45778-y>

The paper does not provide information about the running time or memory requirements of DeepPlant. While performance metrics like accuracy and F1 score are essential, practical factors such as computational efficiency and resource usage are equally important. Reporting details on running time and memory consumption would offer valuable insights into the model's scalability and suitability for large-scale or resource-constrained applications.

Response:

Benchmarking results are indeed crucial for evaluating the usability of the model. We performed benchmarking of DeepPlant on the entire sequencing dataset of *A. thaliana*. The benchmarking results, including runtime and memory consumption, are provided in Supplementary Table 9. We have also added a sentence about the benchmark results in lines 339-341 in the discussion section.

In Figure 3B, it is unclear whether the ratio of profiled sites corresponds to Nanopore coverage or DeepPlant coverage. The authors state that "Reads not aligned to the reference or had low alignment quality were filtered out." From this, it can be inferred that Nanopore coverage is likely higher than DeepPlant coverage. Specifying the coverage obtained for Dorado would also be helpful. Furthermore, the quality threshold for filtering low-quality alignments is not explicitly defined in the manuscript. However, in the code (GitHub commit: 67db8ef; Call_Modification.cpp, lines 107–109), the authors define three constants for filtering, one of which (mapq_thresh_hold) appears to set the mapping quality threshold at 20. More detailed information about the other two heuristics used for filtering is necessary to fully understand the criteria applied during the analysis.

Response:

We recalculated the coverage of DeepPlant and Dorado across different genomic regions and revised Fig. 3b to more clearly illustrate the coverage differences between the three methods. The filtering criteria for low-quality alignments are as follows: (1) MAPQ<20; (2) primary alignment length < 80% of read length; (3) mapping identity <80% . Reads were filtered if less than 80% of the reads were aligned to the . The details have now been described in both the Figure legend in lines 737-739 and in the Methods section in lines 434-436.

Reviewer #1 (Remarks on code availability):

The installation process described on GitHub is problematic and was unsuccessful. Despite having CUDA 11.8 installed and correctly configured (with 'nvcc -version', 'torch.version.cuda', and 'torch.cuda.is_available()' returning the expected output), an error occurs when running cmake. The first line of the error reads: "Could NOT find CUDA (missing: CUDA_INCLUDE_DIRS CUDA_CUDART_LIBRARY) (found version '11.8')." The installation was performed on a server with the 'cuda/11.8.0' and 'cudnn/8.9.2_cu1' module loaded.

Additionally, the installation instructions for CUDA are not user-friendly, as they assume the user has root privileges, which may not always be the case. It's also worth noting that running 'pip install torch==2.0.1' without the '--index-url <https://download.pytorch.org/whl/cu118>' flag installs PyTorch for

CUDA version 11.7. Providing a Dockerfile, a simpler execution option, or clearer setup instructions would improve accessibility and ensure a smoother installation process.

Response:

Thank you for pointing it out. In most cases, the linking location of CUDA in a computer is fixed. However, in some cases, the CUDA path configuration might be inconsistent, leading to compilation errors due to missing libraries. To address potential compilation issues, we provide a Dockerfile along with detailed usage instructions to simplify the use of the tool. The usage instructions are available at <https://github.com/xiaochuanle/DeepPlant>.

Reviewer #2 (Remarks to the Author):

Reviewer #2 (Remarks on code availability):

The installation process on GitHub is problematic and fails despite CUDA 11.8 being correctly configured. Errors occur during cmake, and the instructions assume root privileges, which may not be available to all users. Additionally, installing PyTorch without specific flags defaults to CUDA 11.7, causing potential compatibility issues. Providing a Dockerfile, a simpler installation process, or clearer instructions would greatly improve accessibility and reliability.

Response:

Thank you for your suggestions. In most cases, the linking location of CUDA in a computer is fixed. However, in some cases, the CUDA path configuration might be inconsistent, leading to compilation errors due to missing libraries. To address potential compilation issues, we provide a Dockerfile along with detailed usage instructions to simplify the use of the tool. The usage instructions are available at <https://github.com/xiaochuanle/DeepPlant>.

Reviewer #3 (Remarks to the Author):

Major Revisiong:

The authors developed a deep learning model to process DNA methylome data generated from ONT R10.4 platform. Particularly, they used mCHH enriched genome in the training to improve the accuracy of mCHH data detection. In general, the new model provides

valuable tool for the application of ONT R10.4 in plant epigenome sequencing. I have several concerns:

1)When the authors processed the BS-seq data, how did they treat with the multiple mapped reads? These reads could be generated from repeat enriched regions, such as centromere. if they discard all multiple mapped reads, it will lead to underestimation of mCHH caption in BS-seq data., such as in figure 3b and 3c.

Response:

Thanks for bringing it out. We did not discard all multiple mapped reads. We applied the widely applied Bismark pipeline in BS-seq analysis which only uses unique best alignments in methylation frequency quantification. Unique best alignments refer to alignments where the next best alignment does not exist or is not as good (MAPQ>0), indicating that only multiple mapped reads with multiple best alignments were dumped. For the nanopore data, we also used unique best alignments, except that we required a much higher MAPQ threshold (20) for high alignment accuracy. For causing less confusion, related information has been added in lines 404-405 and 434-436.

2)In figure 3B and 3C, besides BS-seq and Deepplant/Nanopore, the Nanopole data analysis based on Dorodo should also be included to see if there is any improvement using Deepplant then using Dorado.

Response:

Thank you for the suggestion. We have added Dorado's methylation analysis results for the relevant regions, as shown in Figure 3B, Figure 3C, and Supplementary Figures 2 and 5. The manuscript has been revised at related locations.

3)All the analysis are statistical results. The authors should show several screenshots of BS-seq, Dorado, Deepplant data in regions where contain mCG, mCHG and mCHH, and the authors could show these three contexts respectively, to demonstrate that the capture of mCHH is more accurate in Deepplant processed dataset.

Response:

Thank you for the suggestions. We have added methylation analysis results of three methods for other regions, as shown in the Supplementary Figure 2. From these regions, it can be observed that DeepPlant's mCHH analysis results are closer to those of BS-seq, while Dorado's mCHH analysis results in some regions contain a large number of false positives, which should explain the higher correlation coefficient of DeepPlant with BS-seq. The manuscript has been revised accordingly in lines 207-208.

Reviewer #1 (Remarks to the Author):

We appreciate the authors for taking the time to incorporate our suggestions and improve the clarity and completeness of their manuscript. The clear separation of training and evaluation datasets, along with the additional explanations regarding dataset exclusions and public availability, significantly enhance the transparency of the study. The ablation study provides valuable insight into the contribution of individual components, and the inclusion of variability measures strengthens the reliability of the reported results. We also appreciate the clarification on the exclusion of DeepMod2 and the detailed information on DeepPlant's runtime and memory requirements. These thoughtful revisions contribute to a more comprehensive and well-supported analysis.

Concern 1:

A remaining major concern is balanced sampling for read-level evaluation. For evaluation to be truly informative, results should be presented on real-world, naturally imbalanced datasets. Artificially balancing datasets during evaluation does not help users assess a method's practical applicability, as it distorts the class distribution and can inflate performance metrics.

While we agree that some metrics, such as accuracy and AUC-ROC, are not suitable for evaluating highly imbalanced datasets, we cannot agree with the claim that metrics like AP, F1-score, precision, and recall "fail to accurately reflect true performance." These metrics are widely used in the machine learning and bioinformatics communities for precisely this purpose—evaluating models in extreme class imbalance scenarios.

Precision, in particular, can be "unforgiving" when there is a significant imbalance, but this does not make it unrepresentative or biased. Instead, low precision highlights a model's struggle to distinguish true positives from false positives. This is valuable information for users who care about the reliability of positive predictions. Moreover, metrics like average precision (AP) and precision-recall (PR) curves provide an even clearer picture, as they evaluate how well a model ranks positive examples rather than just how it labels them.

In summary, artificially balancing datasets for evaluation risks misleading users about real-world performance. Instead, presenting results on naturally imbalanced datasets—using well-established metrics like precision, recall, F1-score, and AP—ensures transparency and allows users to make informed decisions about method suitability.

Thank you very much for your suggestions. In this revision, we have included single-molecule test results on the imbalanced fully methylated/unmethylated CHH sites across the nine datasets. We agree that evaluating single-molecule performance on realistic, imbalanced datasets can provide valuable insights for researchers. However, the selection of high-quality reference control sites, such as fully methylated (100% methylation frequency) and fully unmethylated (0% methylation

frequency) sites in BS-seq, has been a limiting factor.

Given the scarcity of fully methylated CHH sites in the analyzed species, the ratio of fully methylated to unmethylated CHH sites (Supplementary Table 7) is much lower than the realistic methylation ratios observed in single-molecule data from the same species (Supplementary Table 2). In such cases, single-molecule performance is more effectively indirectly evaluated through methylation frequency quantification. For example, in *A. thaliana*, although the overall CHH methylation content is 3%, the ratio of 100% methylated to 0% methylated sites is 1:32,109. Consequently, direct evaluations of DeepPlant and Dorado on the imbalanced fully methylated/unmethylated datasets primarily provided insights into the accuracy of unmethylated site detection, with DeepPlant outperforming Dorado in all cases (Supplementary Table 7).

Thanks to your valuable suggestions, we have more clearly described the evaluation results and considerations in lines 237-248.

#Concern 2:

We appreciate the authors for providing a detailed description of their method for determining methylation status in a site-level evaluation. While the procedural explanation is now clearer, the specific application of this method remains ambiguous. Is it applied to WGBS data, DeepPlant calls, or both?

Additionally, if this method is applied to DeepPlant, does the term methylation probability refer to the site-level prediction (i.e., the number of methylated Cs divided by the total number of Cs at that site), or does it correspond directly to the model's output? If the threshold is determined using WGBS data, is the same threshold applied to DeepPlant predictions?

Furthermore, if this method is applied to DeepPlant but not Dorado, it would be valuable to evaluate its impact on model performance by applying it to Dorado or by comparing it with a simpler thresholding approach for DeepPlant predictions.

Thank you for your valuable suggestions. Methylation probability refers to a numerical value between 0 and 1, representing the likelihood that a specific CpG motif in a read is methylated, and is therefore different from site-level methylation frequency which is aggregated from the methylation status of multiple reads. As a result, the threshold selection method is only applicable to DeepPlant and Dorado. Following your advice, we assessed the impact of this threshold selection approach on Dorado, and the results showed that the method could also improve the accuracy of Dorado's site-level methylation frequency quantifications, though it only reduced the advantage of DeepPlant by a small proportion. Relevant results are summarized in Table 1. Additionally, we provided a more detailed explanation of this method, which essentially seeks a balance between false positives and false negatives in the model's predictions, thereby reducing the total number of false positives and false negatives as detailed in Supplementary Fig 2. The manuscript has been revised accordingly in lines 203-207.

Minor comments:

We thank the authors for clearly separate training and evaluation datasets. However, some figures could still benefit from having clear separation, e.g. Supplementary Fig. 3

Thank you for pointing out the detail issues in some of the figures. We have separated the datasets in the relevant figures.

Reviewer #1 (Remarks on code availability):

No additional remarks

Reviewer #2 (Remarks to the Author):

Reviewer #2 (Remarks on code availability):

The instructions for running the tool using Docker are clearly written. However, the run command is missing the repository prefix (image name should be given with 'chenhx26/deepplant:1.1.0' instead of just 'deepplant:1.1.0'), which may cause an error when users attempt to run the image.

Thank you for pointing out the issues in the command. We have made corrections to the relevant commands.

Reviewer #3 (Remarks to the Author):

After reviewing the manuscript, I am pleased to suggest that that I have no revisions or objections to raise. The content is well-presented, and it contributes valuable insights to the field. Therefore, I recommend proceeding with its acceptance.

Thank you for considering my feedback. Should you need any further information or discussion, please do not hesitate to contact me.

Reviewer #3 (Remarks on code availability):

Accept